# *Bacillus subtilis* uses the SigM signaling pathway to prioritize the use of its lipid carrier for cell wall synthesis

**Ian J. Roney, David Z. Rudner** *

Department of Microbiology, Harvard Medical School, Boston, Massachusetts, United States of America

* rudner@hms.harvard.edu

**Data Availability Statement:** All relevant data are within the paper and its Supporting Information files.

## Abstract

Peptidoglycan (PG) and most surface glycopolymers and their modifications are built in the cytoplasm on the lipid carrier undecaprenyl phosphate (UndP). These lipid-linked precursors are then flipped across the membrane and polymerized or directly transferred to surface polymers, lipids, or proteins. Despite its essential role in envelope biogenesis, UndP is maintained at low levels in the cytoplasmic membrane. The mechanisms by which bacteria distribute this limited resource among competing pathways is currently unknown. Here, we report that the *Bacillus subtili*s transcription factor SigM and its membrane-anchored anti-sigma factor respond to UndP levels and prioritize its use for the synthesis of the only essential surface polymer, the cell wall. Antibiotics that target virtually every step in PG synthesis activate SigM-directed gene expression, confounding identification of the signal and the logic of this stress-response pathway. Through systematic analyses, we discovered 2 distinct responses to these antibiotics. Drugs that trap UndP, UndP-linked intermediates, or precursors trigger SigM release from the membrane in <2 min, rapidly activating transcription. By contrasts, antibiotics that inhibited cell wall synthesis without directly affecting UndP induce SigM more slowly. We show that activation in the latter case can be explained by the accumulation of UndP-linked wall teichoic acid precursors that cannot be transferred to the PG due to the block in its synthesis. Furthermore, we report that reduction in UndP synthesis rapidly induces SigM, while increasing UndP production can dampen the SigM response. Finally, we show that SigM becomes essential for viability when the availability of UndP is restricted. Altogether, our data support a model in which the SigM pathway functions to homeostatically control UndP usage. When UndP levels are sufficiently high, the anti-sigma factor complex holds SigM inactive. When levels of UndP are reduced, SigM activates genes that increase flux through the PG synthesis pathway, boost UndP recycling, and liberate the lipid carrier from nonessential surface polymer pathways. Analogous homeostatic pathways that prioritize UndP usage are likely to be common in bacteria.

## Introduction

All organisms use polyprenyl-phosphate lipids to transport sugars across membranes [1–3]. In bacteria, the 55-carbon isoprenoid, undecaprenyl phosphate (UndP), ferries a diverse set of

**Funding:** Support for this work comes from the National Institute of Health Grants GM086466, GM127399, GM145299, U19 AI109764 (DZR).

**Competing interests:** The authors have declared that no competing interests exist.

**Abbreviations:** BFP, blue fluorescent protein; CH, casein hydrolysate; LB, lysogeny broth; LTA, lipoteichoic acid; MIC, minimum inhibitory concentration; PG, peptidoglycan; UndP, undecaprenyl phosphate; UndPP, undecaprenyl pyrophosphate; WTA, wall teichoic acid.

sugars and glycopolymers across the cytoplasmic membrane. The most prominent among these is the monomeric building block of the cell wall peptidoglycan (PG). The PG precursor, a disaccharide pentapeptide, is built on UndP in the cytoplasm [4,5]. The lipid-linked muropeptide, called lipid II, is then flipped across the cytoplasmic membrane where it is polymerized and crosslinked into the existing cell wall meshwork. The byproduct of this assembly reaction is undecaprenyl pyrophosphate (UndPP), which is dephosphorylated by membrane phosphatases, and then UndP is flipped back across the cytoplasmic membrane for reuse [6,7]. In addition to PG precursors, UndP transports O-antigen, capsule, exopolysaccharide, and secondary cell wall polymers like wall teichoic acids (WTAs) across the cytoplasmic membrane. The carrier lipid is also used to ferry sugars across the membrane that are used to glycosylate proteins, lipids, and surface polymers. Despite its essential role in these diverse envelope biogenesis and modification pathways, UndP is maintained at low levels in the cytoplasmic membrane (approximately $10^5$ UndP molecules/cell; approximately 0.1% of all membrane lipids) [8]. How cells distribute this limited resource among competing pathways remains an outstanding question in all bacteria. Here, we report that in *Bacillus subtilis*, the alternative sigma factor, SigM, and its anti-sigma factor complex, YhdL-YhdK, function to prioritize UndP for cell wall synthesis.

SigM was first described as a transcription factor required for the outgrowth of *B. subtilis* spores [9]. When spores lacking SigM were induced to germinate in medium containing high salt, the outgrowing cells displayed morphological defects prior to lysis, suggesting they were impaired in cell wall synthesis. SigM mutant cells were subsequently shown to have increased sensitivities to antibiotics that inhibit PG biogenesis [10,11]. Transcriptional profiling experiments revealed that many of the genes in the SigM regulon are involved in cell wall biogenesis [12,13]. Specifically, SigM controls several genes involved in the synthesis and transport of PG precursors as well as PG polymerization and crosslinking. SigM also regulates genes involved in recycling UndP, its de novo synthesis, and liberation of the carrier lipid from UndPP-WTA precursors [14,15]. Work from several groups that spans 2 decades revealed that antibiotics that target virtually every step in PG biogenesis activate SigM-directed gene expression [12,16–18]. A SigM-responsive transcriptional reporter has even been used to screen for small molecules that impair PG synthesis [17,19]. Although responsive to these exogenous stresses, SigM is active at low levels during unperturbed exponential growth, suggesting that its principal function is homeostatic [9].

The *sigM* gene resides in an operon with *yhdL* and *yhdK* that encode integral membrane proteins that hold SigM inactive at the membrane [9,20]. Unlike many membrane anchored anti-sigma factors, the release of SigM is not controlled by regulated proteolysis of its anti-sigma factors [21]. Instead, the YhdL-YhdK (YhdLK) complex is thought to be controlled allosterically. Despite years of study, the signal sensed by the YhdLK complex that triggers release of SigM has remained unclear [13]. Here, we report that antibiotics that trap UndP or UndP-linked intermediates rapidly deplete the UndP pool and trigger SigM release from the membrane within minutes. By contrasts, antibiotics that inhibited cell wall synthesis without directly affecting UndP deplete the carrier lipid more slowly and induce a slower and weaker SigM response. We show this slow response can be explained by sequestration of the carrier lipid in UndP-linked secondary wall polymers. In a complementary set of experiments, we show that depletion of enzymes involved in PG biogenesis mimic the responses observed with antibiotic inhibition. In addition, reduction in UndP synthesis rapidly induces SigM, while increasing UndP production suppresses the SigM response. Importantly, our analysis indicates that UndP-linked precursors do not function as proxies for UndP levels and instead point to UndP as the signal sense by the YhdLK complex. Finally, we show that *sigM* becomes essential when the availability of UndP is limiting. Altogether, our data support a model in which the

YhdLK-SigM pathway functions to prioritize UndP usage for cell wall synthesis. When the levels of free UndP are sufficiently high, the anti-sigma factor complex holds SigM inactive. When levels of the carrier lipid are reduced, SigM is released from the membrane and activates genes that increase PG synthesis, boost UndP recycling, and liberate the lipid carrier from nonessential surface polymer pathways.

## Results

### Antibiotics that block UndP recycling rapidly activate SigM

Previous studies indicate that SigM is activated by virtually all antibiotics that target cell wall synthesis [12,16–18]. The mechanism by which the YhdLK anti-sigma factor complex senses so many distinct blocks to this multistep pathway was unclear (**Fig 1A**). The previous studies on SigM activation were performed by several groups and each analyzed distinct sets of antibiotics using different SigM reporters and assay conditions. As a first step towards defining what the YhdLK-SigM pathway responds to, we systematically tested a large set of antibiotics that target cell wall synthesis under identical conditions. For these experiments, we used a strain harboring a SigM-responsive promoter (P*amj*) fused to *yfp* [22] (**Fig A in S1 Text**) and analyzed SigM activity by fluorescence microscopy. The reporter strain was grown in defined casein hydrolysate (CH) medium to early exponential phase and imaged before and at time points after the addition of each antibiotic at 4 times its minimum inhibitory concentration (MIC).

The SigM response to these antibiotics fell into 2 distinct classes (**Fig 1B and 1D** and **Fig A (C) in S1 Text**). Antibiotics that targeted UndP (amphomycin), UndPP (bacitracin), or UndP-linked cell wall precursors (vancomycin and ramoplanin) activated SigM to high levels within 30 min [23,24]. By contrast, antibiotics that targeted other steps in cell wall synthesis (fosfomycin, penicillin G, D-cycloserine) that do not directly trap UndP activated SigM to lower levels at this time point. Fig 1C shows a more detailed analysis of one antibiotic from each class. Fosfomycin inhibits MurAA and its nonessential paralog MurAB [25,26]. Both enzymes catalyze the first committed step in PG precursor synthesis in the cytosol [25,26]. Vancomycin binds outward-facing lipid II and inhibits the polymerization and crosslinking of PG precursors [27]. Both drugs caused a rapid cessation of growth, indicating they act on similar timescales (**Fig 1C**). However, vancomycin activated SigM to high levels after 30 min, while fosfomycin-treated cells had weak SigM activity at 30 min that increased over time (**Fig 1B**).

To more directly monitor fast- and slow-acting inducers of SigM, we generated a GFP-SigM fusion and monitored its localization before and after antibiotic addition by time-lapse fluorescence microscopy. In the absence of drug, GFP-SigM localized to the membrane in a YhdLK-dependent manner (**Fig 1E** and **Fig B in S1 Text**). Strikingly, addition of drugs that trap UndP (vancomycin, bacitracin, or amphomycin) caused a rapid (1 to 2 min) relocalization of GFP-SigM to the nucleoid (**Fig 1E** and **Fig B(C) in S1 Text**). By contrast, GFP-SigM was membrane associated for >10 min after addition of antibiotics that targeted other steps in cell wall synthesis (**Fig 1E** and **Fig B(C) in S1 Text**).

To correlate SigM activation with the carrier lipid pool, we monitored the free UndP pool before and after antibiotic addition using a fluorescently labeled antibiotic (MX2401) that binds the phosphate moiety on UndP [14,28]. Since MX2401-FL is not membrane permeable, it can only label outward-facing UndP in intact cells. Accordingly, we simultaneously treated cells with MX2401-FL and duramycin, a cyclic peptide that generates pores in the membrane, thereby providing the probe access to both leaflets of the membrane [14,29]. Membrane fluorescence reports on the free carrier lipid pool [14]. Drugs that trap UndP (vancomycin, bacitracin, ramoplanin) caused a rapid (≤2 min) drop in UndP levels (**Fig 1F** and **Fig C in**

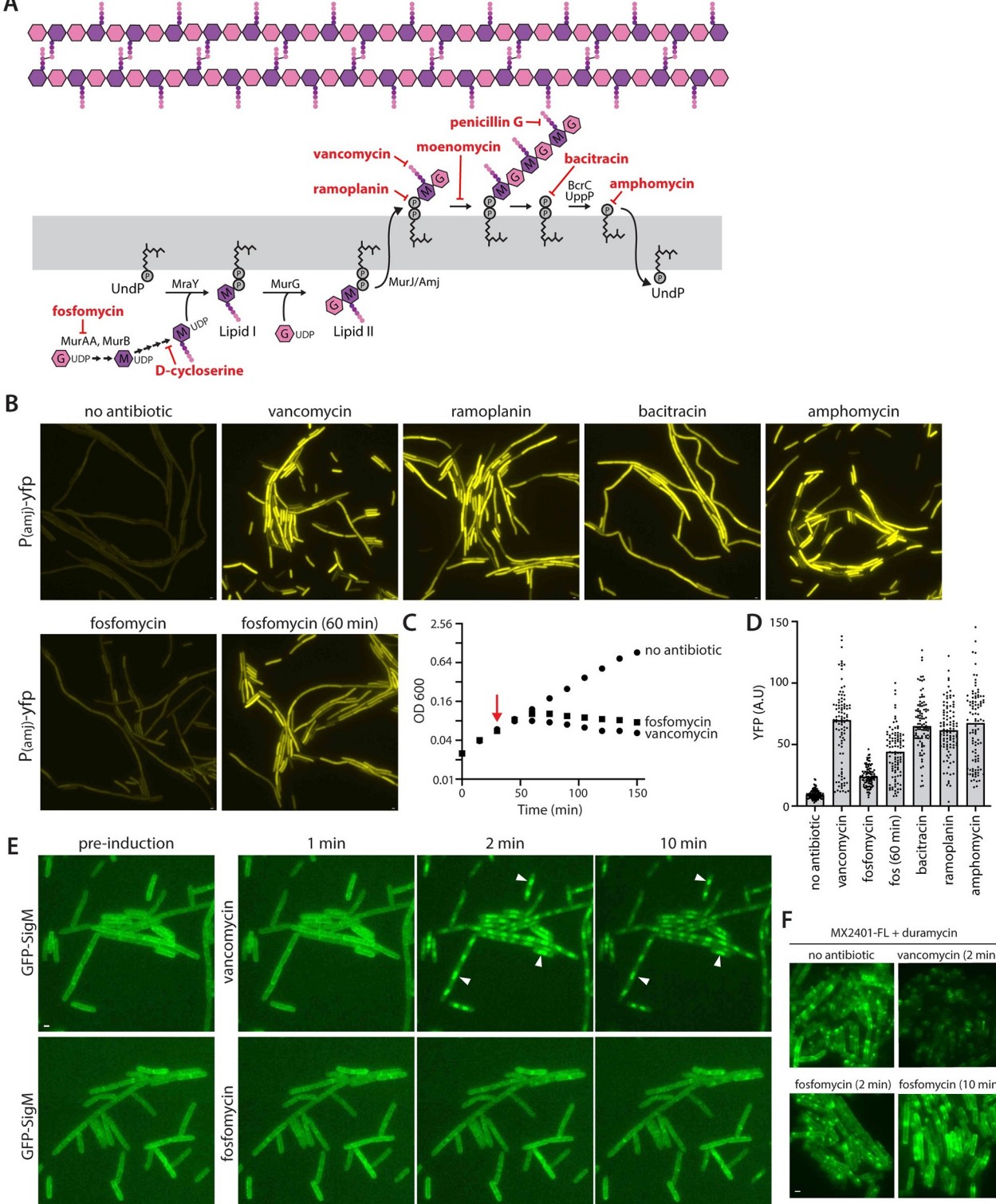

**Fig 1. Antibiotics that block UndP recycling rapidly activate SigM.** (**A**) Schematic of the cell wall synthesis pathway. Antibiotics used in this study are shown in red. Tunicamycin, which targets WTA synthesis, is not shown. Only the enzymes that were depleted in Fig 2 are indicated. (**B**) Representative fluorescence images of *B. subtilis* cells harboring a $\sigma^M$-responsive reporter (P(*amj*)-*yfp*) after exposure to the indicated antibiotics for 30 min. (**C**) Growth curves of wild-type *B. subtilis* treated with vancomycin or fosfomycin. Red arrow indicates the time when antibiotics were added. (**D**) Quantification of fluorescence intensity from images in (**B**). Bar represents median. (**E**) Representative images from time-lapse fluorescence

microscopy of cells expressing a GFP-SigM fusion before and at indicated times after antibiotic exposure. Carets highlight SigM localization to the nucleoid. (**F**) Representative images of wild-type cells treated with the indicated antibiotics for 2 or 10 min and then stained with fluorescently labeled MX2401 (MX2401-FL). Staining was performed in the presence of duramycin, which generates pores in the membrane, allowing MX2401-FL to access inward-facing UndP in addition to outward-facing molecules. Scale bars indicate 1 μm. The data underlying C and D are provided in S1 Data.

S1 Text). By contrast, the carrier lipid pool was largely unchanged after 10 min in the presence of fosfomycin (**Fig 1F**). Thus, the timing of GFP-SigM release from the membrane and its relocalization to the chromosome correlated with the drop in the free pool of UndP.

## Inhibition of WTA synthesis suppresses fosfomycin activation of SigM

Fosfomycin inhibits the first committed step in PG precursor synthesis (**Fig 1A**) and is therefore expected to increase not decrease the pools of UndP. Yet, given sufficient time, the block to precursor synthesis activated SigM. We reasoned that fosfomycin might indirectly deplete the UndP pool by causing an accumulation of UndP-linked WTA precursors. After transport across the cytoplasmic membrane, UndP-WTA polymers are ligated onto nascently synthesized glycan strands (**Fig 2C**) [30,31]. Accordingly, if PG synthesis is blocked due to inhibition of precursor synthesis, outward-facing UndP-WTA should accumulate and deplete the UndP pool. To test this model, we simultaneously treated our reporter strain with fosfomycin and tunicamycin, an inhibitor of the committing step in WTA synthesis [32]. Strikingly, tunicamycin reduced fosfomycin-mediated SigM activation at both 30 and 60 min (**Fig 2A and 2B and Fig D in S1 Text**). Furthermore, UndP levels, as assayed by MX2401-FL, were reduced after exposure to fosfomycin for 20 min, consistent with the observed increase in SigM activity at 30 and 60 min (**Fig 2D and Fig E in S1 Text**). Importantly, tunicamycin largely suppressed the drop in UndP levels caused by fosfomycin (**Fig 2D and Fig E in S1 Text**). Thus, all drugs that target cell wall synthesis directly or indirectly reduce the free pool of the carrier lipid, consistent with the model that the YhdLK-SigM complex is responsive to UndP levels.

## Depletion of cell wall synthesis factors that trap UndP activate SigM

Our data support a model in which SigM is activated by a drop in the pools of UndP and not a general block to cell wall synthesis. To further test this model, we complemented the chemical genetic experiments described above with depletions of enzymes involved in distinct steps in PG biogenesis [4,5]. We generated IPTG-regulated alleles of 3 genes involved in precursor synthesis (*murAA*, *murB*, and *mraY*) that do not directly reduce UndP pools and IPTG-regulated alleles of 3 genes (*murG*, *murJ*, and *bcrC*) that trap UndP-linked intermediates and reduce the pool of the carrier lipid. MurAA and MurB are required for the synthesis of UDP-N-acetyl muramic acid (UDP-MurNAc), while MraY attaches the UDP-MurNAc-pentapeptide onto UndP to generate lipid I (**Fig 1A**). Depletion of MurAA, MurB, or MraY should initially have no impact on UndP or possibly increase the pool of the carrier lipid. MurG catalyzes the conversion of Lipid I to Lipid II and its depletion results in accumulation of lipid I (**Fig 1A**). MurJ and Amj are functionally redundant lipid II flippases that transport the UndP-linked PG precursors from the inner to the outer leaflet of the cytoplasmic membrane [22,33]. Depletion of MurJ in a Δ*amj* strain should accumulate inward-facing lipid II. BcrC and UppP are functionally redundant UndPP phosphatases that regenerate UndP enabling its transport to the inner leaflet of the membrane (**Fig 1A**) [34,35]. Depletion of BcrC in a Δ*uppP* mutant should accumulate outward-facing UndPP.

All strains were precultured in CH medium in the presence of IPTG and then washed and inoculated in medium lacking inducer at low optical density (OD600 = 0.02). Growth was monitored after removal of IPTG and SigM-dependent gene expression was analyzed at the

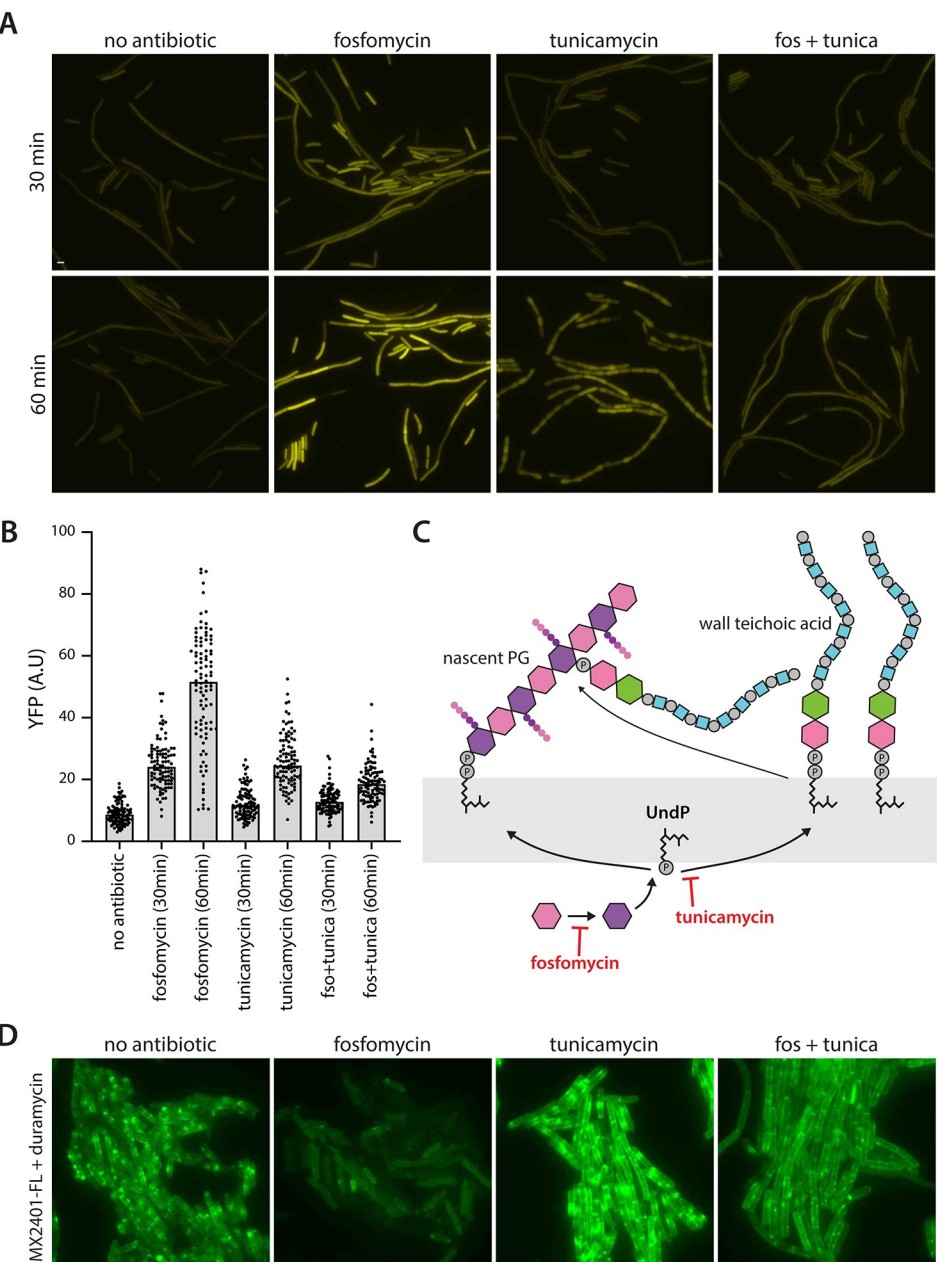

**Fig 2. Inhibition of WTA synthesis suppresses fosfomycin-induced SigM activation.** (**A**) Representative fluorescence images of *B. subtilis* cells harboring a σ^M-responsive reporter (P(*amj*)-*yfp*) after exposure to the indicated antibiotics for 30 or 60 min. (**B**) Quantification of images in (**A**). Bar represents median. (**C**) Schematic illustrating the ligation of an UndP-linked WTA precursor to nascent peptidolgycan (PG). Fosfomycin (fos) and tunicamycin (tunica) inhibit committing steps in PG and WTA synthesis, respectively. (**D**) Representative fluorescence images of wild-type *B. subtilis* cells treated with the indicated antibiotics for 20 min and then stained with fluorescently labeled MX2401 (M2401-FL) that binds UndP. Staining was performed in the presence of duramycin to generate pores in the membrane allowing MX2401-FL to access both inward- and outward-facing UndP. Scale bars indicate 2 μm. The data underlying B are provided in S1 Data.

earliest time point when the strains exhibited a reduction in mass doubling (**Fig 3C**). Importantly, at the time points analyzed a similar percentage of cells had membrane permeability defects as assayed by propidium iodide (**Fig F in S1 Text**), consistent with a similar degree of

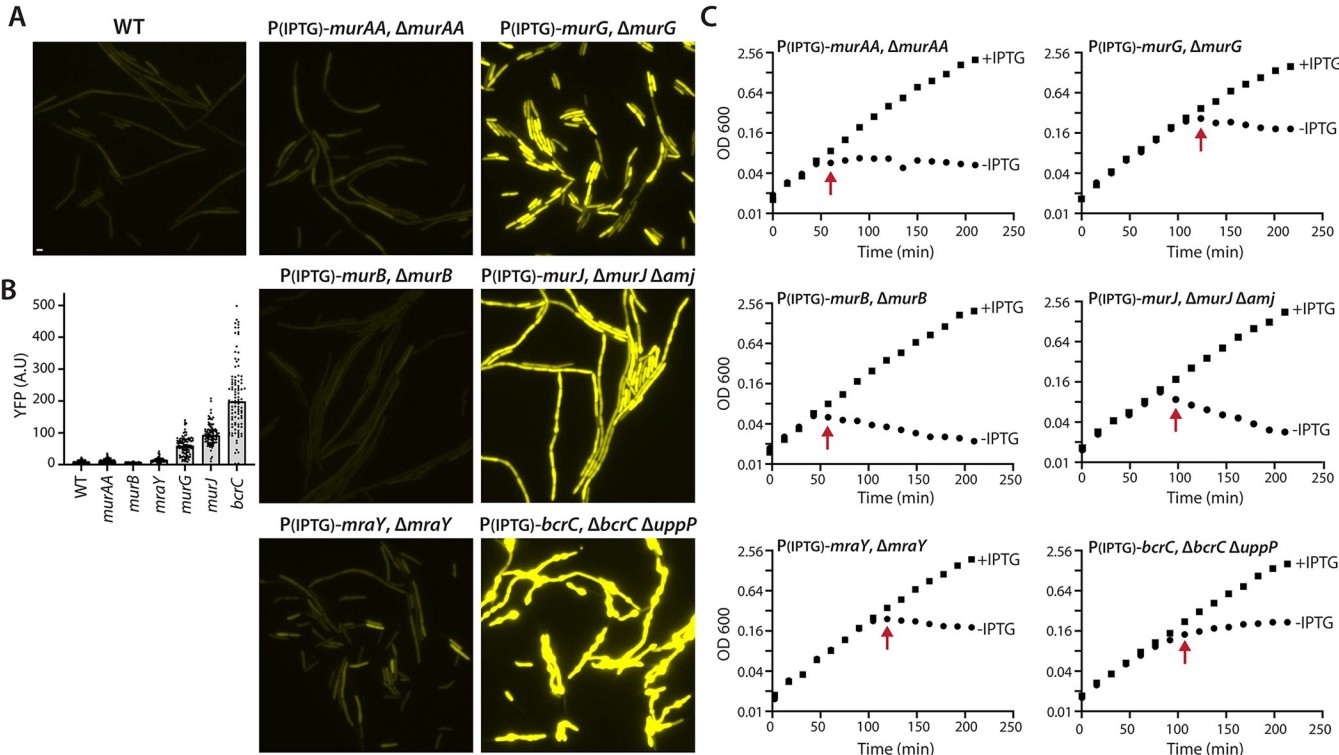

**Fig 3. Depletion of cell wall synthesis enzymes that cause accumulation of UndP-linked precursors activates SigM.** (**A**) Representative fluorescence images of the indicated *B. subtilis* depletion strains harboring a σ$^M$-responsive reporter (P(*amj*)-*yfp*) after removal of IPTG. Scale bar indicates 2 μm. (**B**) Quantification of fluorescence intensity from images in (**A**). Bar represents median. (**C**) Growth curves of depletion strains grown in the presence (squares) or absence (circles) of IPTG. Red arrow indicates the time point at which samples were removed for microscopy in (**A**). IPTG concentrations were *murAA* (100 μM), *murB* (25 μM), *mraY* (25 μM), *murG* (12.5 μM), *murJ* (25 μM), and *bcrC* (25 μM). The data underlying B and C are provided in S1 Data.

cell wall synthesis inhibition. In accordance with our findings with acute exposure to antibiotics, SigM activation upon enzyme depletion fell into 2 classes. Depletion of the enzymes (MurAA, MurB, MraY) that do not directly reduce UndP pools had very weak or undetectable SigM activation, while depletion of the factors (MurG, MurJ, BcrC) that trap UndP strongly activated SigM (>10-fold) (**Fig 3A and 3B**). Importantly all 6 strains had similarly low SigM activity when grown in the presence of IPTG (**Fig G in S1 Text**).

Altogether, the experiments described above indicate that YhdLK-SigM signaling is activated by the accumulation of UndP-linked cell wall precursors and/or a drop in UndP pools rather than a general inhibition of cell wall synthesis. This model is further supported by previous studies showing that depletion of enzymes involved in WTA synthesis that trap UndP-linked WTA intermediates activate SigM [22]. We observed similar results using our transcriptional reporter under the same assay conditions described above. Depletion of the UndP-WTA transporter, TagG, activated SigM to high levels, whereas depletion of the initiating glycosyltransferase, TagO, which does not trap UndP did not (**Fig H in S1 Text**).

## Reducing UndP synthesis rapidly activates SigM

Our data suggest that the YhdLK complex directly monitors UndP levels and activates SigM in response to a drop in the carrier lipid pool. If correct, inhibition of UndP synthesis should strongly activate SigM. A previous study found that a point mutation in the

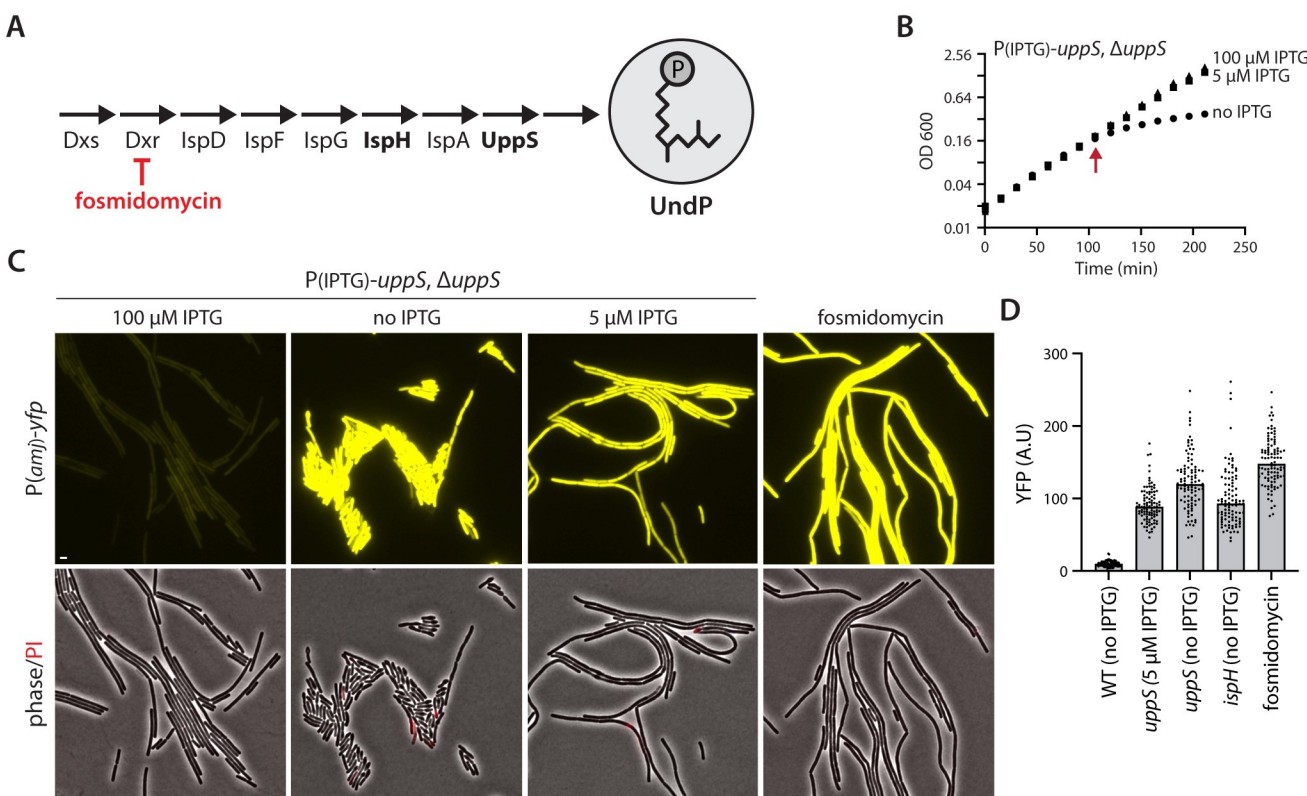

**Fig 4. Depletion of UndP synthesis rapidly activates SigM.** (**A**) Schematic diagram of the UndP synthesis pathway. Fosmidomycin that targets Dxr is highlighted in red. Enzymes (UppS and IspH) that were depleted are in bold. (**B**) Growth curves of UppS depletion strain grown in the presence of 100 μM IPTG (squares), no IPTG (circles), and low (5 μM) IPTG (triangles). Red arrow indicates the time point at which samples were analyzed by fluorescence microscopy. (**C**) Representative fluorescence and phase-contrast images of the *B. subtilis* UppS depletion strain harboring the σ^M-responsive reporter P(*amj*)-*yfp*. Wild-type cells with the same reporter were exposed to fosmidomycin and analyzed by fluorescence microscopy 30 min later. Merged images of phase-contrast and propidium iodide fluorescence are shown below. Scale bar indicates 2 μm. (**D**) Quantification of YFP fluorescence from images like those shown in (**C**). A similar analysis using an IspH depletion strain is presented in Fig I in S1 Text. Quantification of YFP fluorescence from the IspH depletion strain was included in (**D**). The data underlying B and D are provided in S1 Data.

ribosome binding site of *uppS*, the gene encoding UndPP synthase, caused a modest increase in SigM activity [36]. To more directly test whether a reduction in UndP levels impacts SigM activity, we generated IPTG-regulated alleles of 2 essential genes in the UndP biosynthetic pathway (**Fig 4A**), *uppS* and *ispH* [3], and tested how depletion of these enzymes affected SigM activity. Similar to the experiments described above, we monitored growth after removal of IPTG and analyzed SigM activity at the earliest time point when the strains exhibited a reduction in mass doubling. Depletion of either enzyme strongly increased SigM-directed gene expression (**Fig 4 and Fig I in S1 Text**). Furthermore, we identified IPTG concentrations for both depletion strains that did not appreciably impair growth rate but activated SigM to high levels (**Fig 4 and Fig I in S1 Text**). SigM was also strongly activated when UndP synthesis was inhibited by the antibiotic fosmidomycin that targets Dxr [37], an essential enzyme in the isoprenoid biosynthesis pathway (**Fig 4 and Fig I in S1 Text**). We note that fosmidomycin took longer to impair growth compared to the antibiotics that target cell wall synthesis (**Fig I(C) in S1 Text**). This finding is consistent with the idea that the recycled lipid carrier can sustain growth for a period without the production of new UndP molecules to maintain the pool size as the cell elongates.

## UndP-linked sugars and secondary wall polymers are not the signal that activates SigM

Although our results are consistent with a model in which YhdLK directly senses UndP, it was also possible that the complex monitors an UndP-linked polymer or sugar as a proxy for UndP levels. In the context of this model, a reduction in this UndP-linked product rather than UndP itself would cause release of SigM from the YhdLK complex and induction of SigM-controlled gene expression. Depletion of UndP (**Fig 4**) or accumulation of dead-end intermediates in the PG (**Figs 1 and 3**) or WTA (**Fig H in S1 Text**) biosynthesis pathways that all activate SigM would also reduce the synthesis of this UndP-linked product. Accordingly, this alternate model is fully compatible with our data. Since inhibition of the committing steps in the PG and WTA synthesis pathways with fosfomycin and tunicamycin did not activate SigM (**Fig 2 and Fig D in S1 Text**), the hypothetical UndP-linked product that is monitored by YhdLK cannot be an UndP-linked intermediate in the PG or WTA synthesis pathways. However, UndP is also used for the synthesis of a minor WTA polymer, an anionic polymer with reduced phosphate called teichuronic acid, and the glycosylation of lipoteichoic acid (LTA) (**Fig 5A**) [38–41]. In addition, there are 2 less well-characterized cell surface glycosylation pathways that are thought to use UndP as a lipid carrier [39].

The minor WTA and teichuronic acid polymers are not essential. Accordingly, we deleted the genes (*tua* and *ggaA*) encoding the committing enzymes in these pathways and monitored SigM activity. As can be seen in Fig 5B, SigM activity remained low in the mutants, indicating UndP-linked intermediates in these pathways are not monitored by YhdLK. Similarly, none of the surface glycosylation pathways that use UndP are essential and deletion of all 3 genes (*csbB*, *ykoT*, and *ykkC*) that transfer the sugar to UndP failed to activate SigM (**Fig 5B**). Importantly, addition of vancomycin to these deletion strains induced SigM activity, indicating they are still capable of generating the signal sensed by YhdLK (**Fig 5B**). Some *B. subtilis* strains produce exopolysaccharides built on UndP. However, the strain we are using, PY79, does not [42]. Altogether, these data indicate that YhdLK does not monitor an UndP-linked polymer or sugar as a proxy for UndP. Instead, these data argue that the anti-sigma factor complex directly monitors the pool of free UndP. When UndP levels are reduced, SigM is released from the membrane and activates the genes under its control.

We used these sugar modification and secondary wall polymer synthesis pathways to further explore our model. It was previously reported that deletion of *yfhO*, encoding the ligase that transfers UndP-GlcNAc to LTA, activates SigM [43]. In the context of our model, SigM activation results from the accumulation of unligated UndP-GlcNAc and the sequestration of UndP. We extended these findings by generating IPTG-regulated alleles of *csbB*, *ykcC*, and *ykoT* that encode the glycosyltransferases that generate UndP-linked sugars (**Fig 6C and Fig J (B) in S1 Text**). We inserted these regulated alleles into strains lacking the *csbB-yfhO*, *ykcC-ykcB*, or *ykoS-ykoT* operons, respectively. Since YfhO, YkcB, and YkoS transfer the UndP-linked sugar onto surface polymers (**Fig 6C and Fig J(B) in S1 Text**), overexpression of the glycosyltransferases that commit UndP should rapidly trap UndP-linked sugars and activate SigM. In the absence of IPTG, these strains had low SigM activity (**Fig 6A and 6B and Fig J in S1 Text**). However, 60 min after IPTG addition, SigM activity was induced to high levels, and, in the case of CsbB and YkcC overexpression, the cells began to bulge, consistent with impaired PG and WTA synthesis (**Fig 6A and 6B and Fig J in S1 Text**) [43]. SigM activity was not as strongly induced in the cells overexpressing YkoT nor did the cells display morphological defects, suggesting that YkoT was not as active or highly expressed (**Fig J in S1 Text**). Finally, we observed similar results when we overexpressed the committing enzymes (GgaA and TuaA) in the minor WTA or teichuronic acid synthesis pathways in strains blocked at a

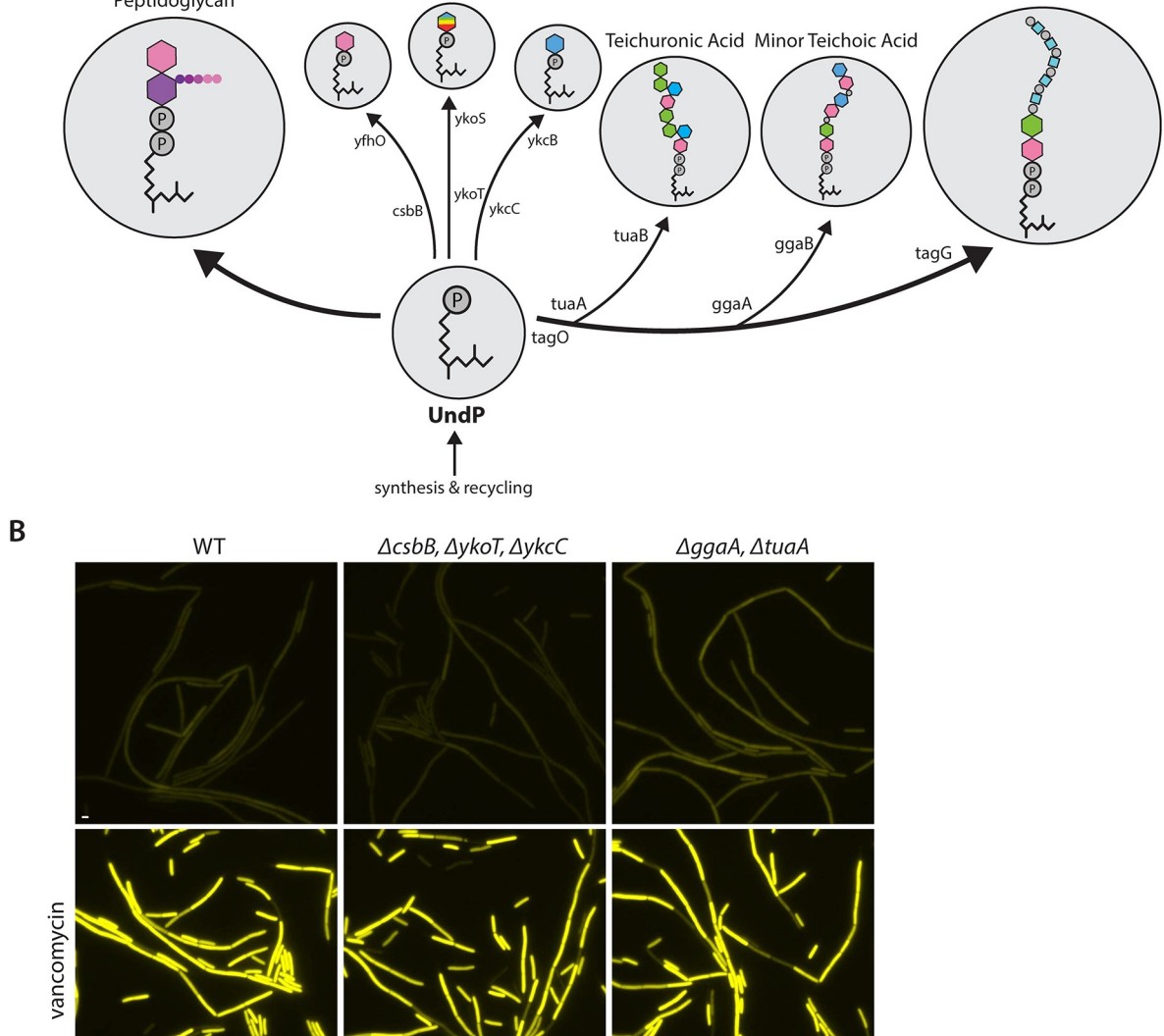

**Fig 5. Reduced levels of UndP-linked sugars, minor teichoic acid, or teichuronic acid are not the signal that activates SigM signaling.**
(**A**) Schematic of the *B. subtilis* biosynthetic pathways that use UndP as a carrier lipid. (**B**) Representative fluorescence images of the indicated *B. subtilis* strains harboring the σ$^{M}$-responsive reporter P(*amj*)-*yfp*. The absence of all 3 UndP-linked sugars involved in surface polymer glycosylation has no impact on SigM activity. The absence of teichuronic acid and the minor WTA similarly had no impact on SigM activity. The mutants activate SigM upon exposure to vancomycin, indicating they are still responsive to an activating signal. Scale bar indicates 2 μm.

downstream step (Δ*ggaB* or Δ*tuaB*, respectively), trapping UndP-linked intermediates (**Fig 6A and 6B** and Fig K and L in **S1 Text**).

The experiments in Fig 2 show that tunicamycin-inhibition of the committing step in WTA synthesis suppressed SigM activation caused by fosfomycin-inhibition of the committing step in PG synthesis. We interpreted this suppression as evidence that fosfomycin induces SigM because UndP-linked WTA precursors accumulate in the absence of de novo PG synthesis, reducing the UndP pool. Our model predicts that the overexpression of CsbB in the absence of YfhO activates SigM due to sequestration of UndP in UndP-GlcNac. If correct, the addition of

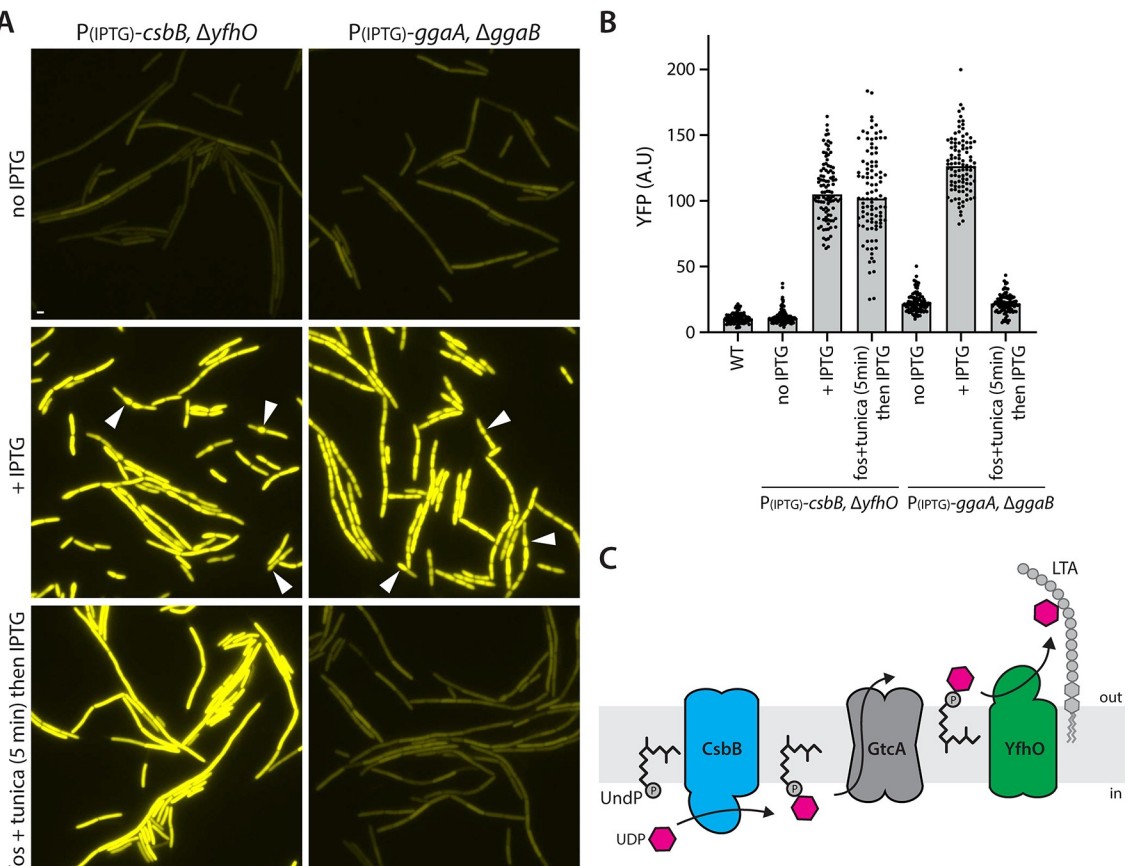

**Fig 6. Trapping UndP in surface polymer modification pathways activates SigM.** (**A**) Representative fluorescence images of the indicated *B. subtilis* strains harboring the σ^M-responsive reporter P(*amj*)-*yfp*. Overexpression of CsbB in the absence of YfhO traps UndP-GlcNAc and activates SigM. Pretreatment with fosfomycin (fos) and tunicamycin (tunica) prior to IPTG addition does not suppress SigM activation. Overexpression of GgaA in the absence of GgaB traps UndP-linked teichoic acid precursors and activates SigM. Pretreatment with fos and tunica blocks precursor synthesis and SigM activation. Carets highlight cells with morphological defects. Scale bar indicates 2 μm. (**B**) Quantification of fluorescence images from (**A**). Bar represents median. (**C**) Schematic of the LTA glycosylation pathway. The data underlying B are provided in S1 Data.

tunicamycin and fosfomycin during overexpression of CsbB should have no impact on the activation of SigM. This was indeed the case (**Fig 6 and Fig J in S1 Text**). By contrast, the minor teichoic acid is built on the WTA linkage unit (**Fig L in S1 Text**), and, therefore, inhibition of the committing step in WTA synthesis by tunicamycin is predicted to suppress SigM activation [38,40]. As can be seen in Fig 6, tunicamycin effectively suppressed SigM activation. Collectively, these experiments provide additional support for the model that SigM is released from the YhdLK membrane complex when the pools of UndP are reduced.

## Defects in LTA biogenesis weakly activate SigM and activation is suppressed by UppS overexpression

Mutations in the LTA synthesis pathway have been reported to activate SigM [44,45]. LTA synthesis is the only surface polymer in *B. subtilis* that is not built on UndP [46]. Accordingly, SigM activation in LTA synthesis mutants appears inconsistent with the model that YhdLK responds to changes in UndP [13]. We therefore revisited the LTA synthesis mutants using our SigM reporter and assay conditions. Specifically, we generated deletions of *ugtP* and *ltaS*

encoding 2 enzymes involved in LTA synthesis [46]. UgtP is required for the synthesis of LTA's glucolipid anchor, and LtaS is the primary LTA synthase. Mutations in these genes were previously reported to activate SigM. However, under our assay conditions, cells harboring the ΔugtP mutation did not activate SigM while the ΔltaS mutation increased SigM activity approximately 3-fold (**Fig M in S1 Text**). To investigate whether the differences were due to the growth medium used, we repeated the experiments in LB medium as was used previously [44]. Under these conditions, both the ΔugtP and ΔltaS mutations increased SigM activity by approximately 2-fold. For comparison, perturbations that depleted UndP or trapped UndP-linked precursors induced SigM activity by approximately 10- to 50-fold. We therefore suspected that the absence of UgtP or LtaS was indirectly reducing the carrier lipid pool. To investigate this possibility, we tested whether overexpression of UppS could suppress the SigM activation in the mutants. As can be seen in Fig M in S1 Text, UppS overexpression largely suppressed the increase in SigM activity resulting from defects in LTA synthesis. These findings provide further support for the model that the YhdLK complex is responsive to UndP levels.

## SigM becomes essential when UndP is limiting

Many of the genes under SigM control are involved in cell wall biogenesis including synthesis and transport of PG precursors (*ddl*, *murB*, *murF*, *amj*) and PG polymerization and crosslinking (*rodA*, *ponA*, *mreB*, *mreC*, *mreD*, *divIC*, *divIB*) [6,12,13]. Furthermore, SigM controls genes involved in UndP recycling (*bcrC*, *uptA*) [10,14], de novo synthesis (*ispD*, *ispF*) [3], and liberation of the carrier lipid from UndPP-WTA precursors (*tagT*, *tagU*) [15]. We hypothesize that the increase in SigM-directed gene expression when the carrier lipid pool is low allows the cell to prioritize PG synthesis over other surface polymer or modification pathways while simultaneously increasing the pool of UndP (**Fig 7A**). This model predicts that SigM will become essential when the synthesis or recycling of UndP is decreased. To test this, we compared plating efficiency using spot-dilutions of strains with and without *sigM* under conditions in which synthesis or recycling of UndP was reduced (**Fig 7B**). Specifically, we analyzed IPTG-regulated alleles of *uppS*, *ispH*, or *bcrC*. In the presence of 500 μM IPTG, all strains grew similarly. However, with lower concentrations of IPTG, the strains lacking *sigM* had severe plating defects or failed to form colonies (**Fig 7B**). Similar results were obtained when we depleted TuaB that transports UndP-linked teichuronic acid [41]. When expression of TuaB was reduced, UndP-link teichuronic acid precursors accumulate and SigM became essential (**Fig 7B**). Furthermore, overexpression of CsbB, YkcC, or YkoT in strains lacking their cognate ligase also required SigM for viability (**Fig 7C**). Collectively, these data argue that SigM activation maintains viability by boosting PG synthesis and UndP recycling when the pools of the carrier lipid become limiting.

## Discussion

Altogether, our data support a model in which the YhdLK complex monitors UndP and, when levels of the carrier lipid are low, releases SigM from the membrane triggering the activation of genes that increase flux through the cell wall biogenesis pathway, UndP synthesis, recycling, and liberation (**Fig 7A**). Although we have not provided direct evidence that UndP is sensed by YhdLK, this model represents the simplest explanation for the entire body of data presented. It can explain the large number of distinct perturbations that activate SigM; the rapidity by which SigM is released from the membrane upon exposure to antibiotics that trap UndP and UndP-linked intermediates and precursors; the close correlation between the reduction in the UndP pool and SigM release from the membrane; and the suppression of SigM activation

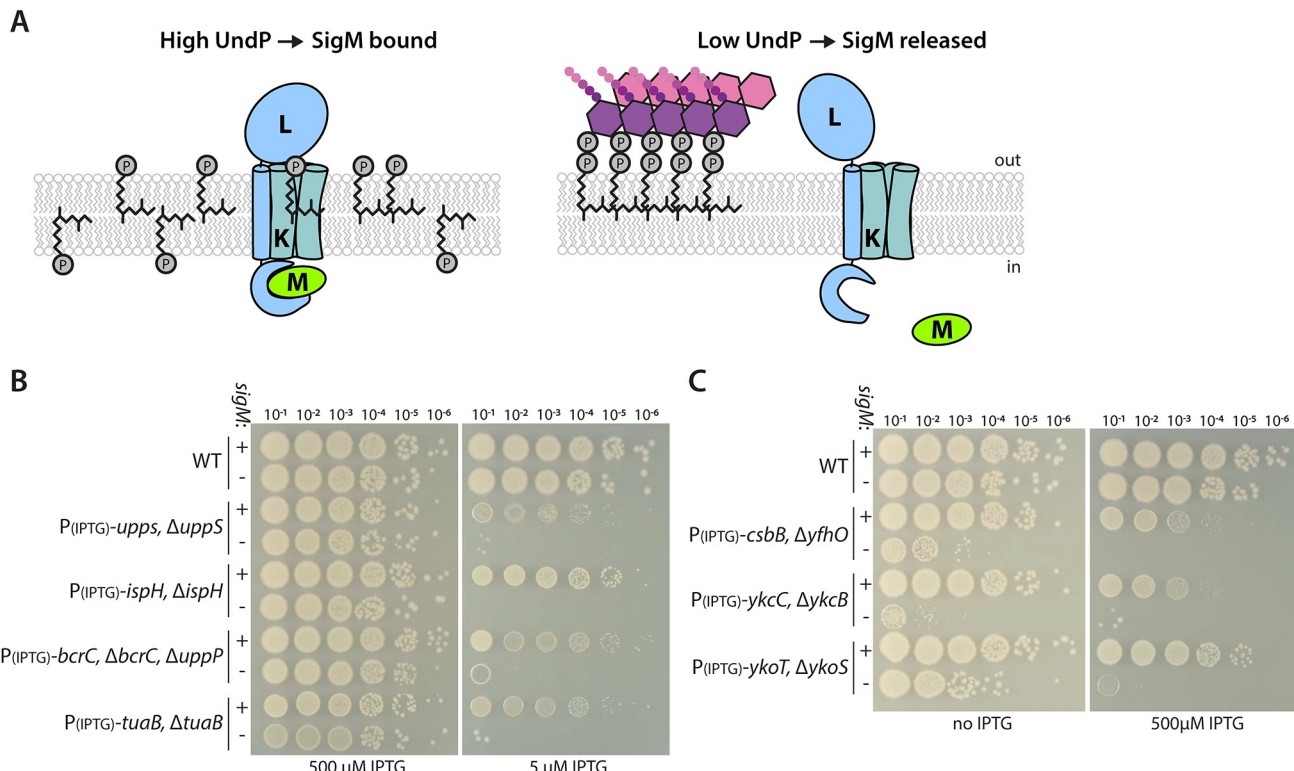

**Fig 7. SigM becomes essential when free UndP is limited.** (**A**) Schematic model of the regulation SigM activity. YhdL (L) and YhdK (K) hold SigM (M) inactive at the membrane when UndP levels are high (Left) and release SigM when UndP levels drop (Right). In this example, UndP levels drop due to its sequestration in lipid II. (**B**) Spot-dilution assays of the indicated depletion strains in the presence and absence of SigM. Under permissive (500 μM IPTG) conditions, sigM is not essential. Under partially restrictive (5 μM IPTG) conditions, UndP becomes limiting and sigM is essential for viability. (**C**) Spot-dilution assays of the indicated overexpression strains in the presence and absence of SigM. sigM is essential when UndP is trapped in unproductive pathways.

by increased production of UndP. Finally, we have ruled out all potential proxies for UndP. We propose that binding of UndP to the YhdLK complex stabilizes the interaction between the membrane-anchored anti-sigma factor and SigM. When the UndP pools are reduced, binding to YhdLK is lost and SigM is released (**Fig 7A**). It is noteworthy that the activating step in this signaling pathway is the loss or reduction of a signal (UndP) rather than the appearance of one. The *Streptomyces* sigma factor Sig^WhiG and its anti-sigma factor RsiG represent another example in which the loss or reduction of a signal (cyclic-di-GMP) relieves inhibition of a sigma factor [47]. We suspect this mode of regulation of alternative sigma factors is more common than currently appreciated.

## A homeostatic pathway for UndP usage

Although our characterization of SigM activation utilized chemical and genetic perturbations, SigM is active during unperturbed growth [9]. Accordingly, we envision that SigM principally functions to homeostatically control UndP usage. Small changes in UndP levels are likely to cause subtle changes in the active pool of SigM, thereby modulating flux through the PG biogenesis pathway. Thus, the YhdLK/SigM pathway serves to maintain PG synthesis in the face of these changes.

An alternative mechanism to prioritize UndP usage is to simply increase de novo synthesis of UndP when the levels of the carrier lipid drop. However, we found that increasing

expression of the UndP synthase, UppS, caused impaired growth and morphological defects (**Fig N in S1 Text**). It is not clear whether higher levels of UndP impacts the lipid bilayer or alters the synthesis of surface polymers. Either way, these findings provide an explanation for why *B. subtilis* has evolved a mechanism to enhance UndP recycling and prioritize the available pool for PG synthesis. Although SigM controls 2 genes in the isoprenoid biosynthesis pathway (*ispD* and *ispF*), based on our UppS overexpression data, we hypothesize that their induction has a modest impact on de novo synthesis of UndP.

The YhdLK-SigM pathway is confined to *B. subtilis* and close relatives of the genus *Bacillus*. Nonetheless, our data suggest that other bacteria have evolved analogous pathways to distribute the lipid carrier among competing pathways. The identities of these pathways await discovery in the future.

## Cell envelope homeostatic pathways in *B. subtilis*

The YhdLK-SigM pathway is one of at least 4 signal transduction pathways in *B. subtilis* dedicated to cell envelope homeostasis. The WalR-WalK 2-component signaling system functions to modulate cell wall hydrolases [48]. The sensor kinase WalK monitors the activity of D,L-endopeptidase required for expansion of the cell wall during growth and modulates their levels and activities in response [49,50]. The SigI-RsgI sigma factor/anti-sigma factor pair monitors the cell wall for defects in the meshwork and activates genes that promote its fortification [51,52]. Meanwhile, the serine/threonine kinase PrkC monitors some aspect of cell wall synthesis, possibly the UndP-linked PG precursor Lipid II [53], and modulates the activities of several proteins involved in cell wall biogenesis via phosphorylation [54,55]. Finally, here, we have shown that the SigM-YhdLK pathway monitors the available pool of UndP and prioritizes PG synthesis over other surface polymer and modification pathways. The goal for the future is to understand the logic of sensing and responding to these specific inputs and to dissect how these pathways interface with each other to maintain robust cell envelope synthesis during growth.

## SigM signaling as a tool for antibiotic discovery

Although SigM would not be a good antibiotic target, the findings presented here reinforce the idea that small molecule screens based on the activation of a SigM-responsive reporter are a worthwhile strategy to identify compounds that impair virtually all steps in envelope biogenesis. In fact, a SigM activity reporter has been used previously to screen for small molecules that target the cell wall [17,56]. The hits from this screen were found to specifically target UndP synthesis. Even weak hits from similar screens have the potential to identify novel inhibitors of cell wall synthesis that trap UndP intermediates or impair precursor biogenesis. These could serve as the starting point for medicinal chemistry campaigns to improve potency and selectivity.

## Methods

### General methods

All *B. subtilis* strains were derived from the prototrophic strain PY79 [54]. All *B. subtilis* experiments were performed at 37°C with aeration in defined CH medium [55] or lysogeny broth (LB). Antibiotic concentrations used were 100 μg/mL spectinomycin, 10 μg/mL kanamycin, 5 μg/mL chloramphenicol, 10 μg/mL tetracycline, 1 μg/mL erythromycin, and 25 μg/mL lincomycin (MLS). All *B. subtilis* strains were generated using the 1-step competence method unless indicated otherwise. All strains, plasmids, and oligonucleotides used in this study can be found

in Tables A-C in S1 Text. All experiments presented are representative of at least 3 biological replicates.

## Growth curve assays

Depletion strains were grown in CH medium under permissive conditions until late log phase. Permissive IPTG concentrations for the depletions strains were *murA* (100 μM), *murB* (25 μM), *mraY* (25 μM), *murG* (12.5 μM), *murJ* (25 μM), *bcrC* (25 μM), *uppS* (100 μM), *ispH* (25 μM), *tuaB* (500 μM), *tagO* (12.5 μM), and *tagG* (100 μM). Cultures were pelleted at 4,000 ×*g* for 5 min and washed once with 25 mL of CH medium. Cultures were normalized to a starting OD of 0.05 in 25 mL CH medium with or without IPTG in 250 mL baffled flasks. Cultures were grown for 3.5 h with shaking at 37˚C, and samples were taken every 15 min for OD600 measurements. Samples were analyzed by fluorescence microscopy at the indicated time points. For depletion experiments, samples were analyzed by microscopy at the earliest time point at which the depletion strain had a discernable growth defect as measured by OD600 in comparison to the permissive condition. Growth curves were performed at least 3 independent times, and representative graphs plotted with GraphPad Prism are shown. For *uppS* overexpression experiments, cultures were grown in CH medium under permissive conditions until late log phase. Cultures were back-diluted to a starting OD of 0.05 in 25 mL CH medium with the indicated concentration of IPTG in 250 mL baffled flasks.

## Spot-dilution assays

*B. subtilis* strains were grown at 37˚C with aeration in LB until cultures approached late-log phase. Cultures were normalized to OD600 = 1, and 10-fold serial dilutions were generated. Approximately 5 μL of each dilution were spotted onto LB agar supplemented with or without indicated concentrations of IPTG. Plates were incubated at 37˚C overnight and photographed the next day.

## Antibiotic induction experiments

*B. subtilis* strains were grown at 37˚C with aeration in CH medium until cultures approached mid-log phase (OD600 = 0.2). Antibiotics were added at 4×MIC, and samples were analyzed by fluorescence microscopy 30 and 60 min later. Antibiotic concentrations used were vancomycin (2 μg/mL), fosfomycin (200 μg/mL), Penicillin-G (40 μg/mL), bacitracin (1 mg/mL), amphomycin (20 μg/mL), tunicamycin (32 μg/mL), and fosmidomycin (12.5 μg/mL).

## Glycosyltransferase overexpression experiments

*B. subtilis* strains harboring IPTG-regulated alleles of *csbB*, *ykoS*, or *ykoT* were grown at 37˚C with aeration in CH medium until cultures approached mid-log phase (OD600 = 0.2). IPTG was added to a final concentration of 500 μM, and samples were analyzed by fluorescence microscopy 60 min later. When fosfomycin and tunicamycin were used, they were added to the culture 5 min before IPTG addition.

## Fluorescence microscopy

Cells were washed with 1×PBS, resuspended in 1/25 volume of 1×PBS, and spotted onto 1.5% agarose pads containing growth medium. Propidium iodide labeling was performed in 1×PBS at final concentrations of 5 μM.

Phase and fluorescence microscopy was performed with a Nikon Ti inverted microscope using a Plan Apo 100×/1.4 Oil Ph3 DM objective, a Lumencore SpectraX LED illumination

system and an Andor Zyla 4.2 Plus sCMOS camera. Chroma ET filter cubes (#49000, 49002, 49003, and 49008) were used for imaging BFP, sfGFP, YFP, and propidium iodide. Exposure times were 50 ms (prodium iodide); 200 ms (sfGFP); 200 ms (MX2401-FL), 400 ms (BFP); and 1 s (YFP). Images were acquired with Nikon elements 4.3 software and analyzed using ImageJ (version2.3).

## GFP-SigM localization by time-lapse fluorescence microscopy

BIR1100 [*ycgO*::*Pspank-LK(spec)*, Δ*mlk*::*erm*, *yvbJ*::*PxylA-sfGFP-sigM(kan)*] was grown at 37˚C with aeration in CH broth containing 0.3 mM xylose and 500 μM IPTG until cultures approached mid-log phase. Cells were spotted onto coverslips and covered with 1.5% agarose pads containing CH medium. Coverslips were mounted on the microscope, and images were taken every minute for 10 min. After the first image was taken, 5 μL of the indicated antibiotics was pipetted onto the top of the agarose pad. Antibiotics were used at high concentration to ensure rapid diffusion through the pad and inhibition of cell wall synthesis. Images shown are representative of at least 3 biological replicates.

## Fluorescence microscopy quantification

ImageJ was used to quantify fluorescent intensities. For quantification of P(*amj*)-*yfp*, a vegetatively expressed blue fluorescent protein (BFP) was used to identify cell boundaries. Intensity values from the YFP channel were extracted and the background autofluorescence from an empty field of view was subtracted from the image. Fluorescence intensities from 100 cells from multiple fields were plotted using GraphPad Prism 9.

## MX2401-FL labeling and fluorescence microscopy

Exponentially growing cultures of *B. subtilis* were collected by centrifugation at 7,000 RPM for 2 min. Cells were washed once with $1 \times$PBS + 25 μg/mL CaCl$_2$ (pH 7.4) and resuspended in 1/25th volume of $1 \times$PBS + 25 μg/mL CaCl$_2$. A mixture of MX2401 fluorescently labeled with CF488A (Biotium #92350) (MX2401-FL) [14] (25 μM final) and duramycin (25 μg/mL final), which generates pores in the membrane allowing MX-FL access to the cytoplasmic-facing UndP, was added and incubated for 60 s. Cells were washed with $1 \times$PBS and spotted onto 1.5% agarose pads containing growth medium. Propidium iodide (5 μM final) was added to monitor permeabilization by duramycin.

## *B. subtilis* deletion mutants

Most *B. subtilis* deletion mutants were made by isothermal assembly [57] followed by direct transformation. The assembly reactions contained 3 PCR products: Two of the products were approximately 1,500 base pairs upstream and downstream of the gene to be deleted, and the third product contained an antibiotic resistance cassette. Antibiotic resistance cassettes flanked by lox66/lox71 sites were amplified from pWX465(cat), pWX466(spec), pWX467(erm), pWX469(tet), and pWX470(kan) using the primers oJM028 and oJM029. The flanking regions for the respective deletions were amplified using PY79 genomic DNA as template and the following primer sets: *murAA(oIR423-426)*, *murB(oIR427-430)*, *mraY(oIR344-347)*, *murG (oIR388-391)*, *uppP(oIR419-422)*, *tagO(oIR340-343)*, *tagG(oIR384-387)*, *sigM-yhdL-yhdK(oIR)*, *ykoST(oIR765-768)*, *csbB-yfhO(oIR769,770,078,079)*, *ykcBC(oIR761-764)*, *ggaAB (oIR710,711,716,717)*, *ugtP(oIR747-750)*. The *bcrC* and *ltaS* deletions were from the BKE collection and were backcrossed twice into PY79 and PCR confirmed. The *amj* and *murJ* deletions were previously described [22].

## Plasmid constructions

**pGD024 [ycgO::Phyperspank-uppS (spec)(amp)].** pGD024 was generated in a 2-piece isothermal assembly reaction with a PCR product containing the *uppS* gene (amplified from PY79 gDNA with oGD87 and oGD88) and pCB090 [ycgO::Phyperspank(spec)] digested with NheI and HindIII.

**pIR086 [ycgO::Pspank-yhdLK (spec)(amp)].** pIR086 was generated in a 2-piece isothermal assembly reaction with a PCR product containing the *yhdL* and *yhdK* genes (amplified from PY79 gDNA with oIR153 and oIR159) and pCB084 [ycgO::Pspank(spec)] digested with HindIII and XmaI.

**pIR175[ycgO::Phyperspank-csbB(spec)(amp)].** pIR175 was generated in a 2-piece ligation with a PCR product containing the *csbB* gene (amplified from PY79 gDNA with oIR338 and oIR339) and pCB090 [ycgO::Phyperspank(spec)] digested with HindIII and SpeI.

**pIR176[ycgO::Pspank-mraY(spec)(amp)].** pIR176 was generated in a 2-piece ligation with a PCR product containing the *mraY* gene (amplified from PY79 gDNA with oIR336 and oIR337) and pCB084 [ycgO::Pspank(spec)] digested with HindIII and SpeI.

**pIR177[ycgO::Pspank-tagO(spec)(amp)].** pIR177 was generated in a 2-piece ligation with a PCR product containing the *tagO* gene (amplified from PY79 gDNA with oIR334 and oIR335) and pCB084 [ycgO::Pspank(spec)] digested with HindIII and SpeI.

**pIR190[ycgO::Pspank-tagG(spec)(amp)].** pIR190 was generated in a 2-piece isothermal assembly reaction with a PCR product containing the *tagG* genes (amplified from PY79 gDNA with oIR372 and oIR373) and pCB084 [ycgO::Pspank(spec)] digested with HIndIII and SpeI.

**pIR192[ycgO::Pspank-murJ(spec)(amp)].** pIR192 was generated in a 2-piece ligation with an insert containing the *murJ* gene (digested from pAM133 with EcoRI and BamHI) and pCB084 [ycgO::Pspank(spec)] digested with EcoRI and BamHI.

**pIR193[ycgO::Pspank-murG(spec)(amp)].** pIR193 was generated in a 2-piece ligation with an insert containing the *murG* gene (digested from pAM125 with HindIII and SphI) and pCB084 [ycgO::Pspank(spec)] digested with HindIII and SphI.

**pIR194[ycgO::Pspank-bcrC(spec)(amp)].** pIR194 was generated in a 2-piece ligation with PCR product containing the *bcrC* gene (amplified from PY79 gDNA with oIR374 and oIR375) and pCB084 [ycgO::Pspank(spec)] digested with HindIII and SpeI.

**pIR209[ycgO::Pspank-murAA(spec)(amp)].** pIR209 was generated in a 2-piece ligation with a PCR product containing the *murAA* gene (amplified from PY79 gDNA with oIR435 and oIR436) and pCB084 [ycgO::Pspank(spec)] digested with HindIII and SpeI.

**pIR210[ycgO::Pspank-murB(spec)(amp)].** pIR210 was generated in a 2-piece ligation with PCR product containing the *murB* gene (amplified from PY79 gDNA with oIR437 and oIR438) and pCB084 [ycgO::Pspank(spec)] digested with HindIII and SpeI.

**pIR286[ycgO::Phyperspank-ykoT(spec)(amp)].** pIR286 was generated in a 2-piece ligation with PCR product containing the *ykoT* gene (amplified from PY79 gDNA with oIR783 and oIR784) and pCB090 [ycgO::Phyperspank(spec)] digested with HindIII and SpeI.

**pIR287[ycgO::Phyperspank-ykcC(spec)(amp)].** pIR287 was generated in a 2-piece ligation with a PCR product containing the *ykcC* gene (amplified from PY79 gDNA with oIR785 and oIR786) and pCB090 [ycgO::Phyperspank(spec)] digested with HindIII and SpeI.

**pIR288[ycgO::Phyperspank-ggaA(spec)(amp)].** pIR288 was generated in a 2-piece isothermal assembly reaction with a PCR product containing the *ggaA* gene (amplified from PY79 gDNA with oIR787 and oIR788) and pCB090 [ycgO::Phyperspank(spec)] digested with HindIII and SpeI.

**pIR344[yvbJ::PxylA-sfGFP-sigM(kan)(amp)].** pIR344 was generated in a 3-piece isothermal assembly reaction with PCR products containing the *sfGFP* gene (amplified from

pIR328 with oIR657 and oIR697), *sigM* (amplified from PY79 gDNA with oIR961 and oIR962) and pCB133 [yvbJ::PxylA(kan)] digested with XhoI and BamHI.

All plasmids were confirmed by sequencing.

## Supporting information

**S1 Data. Spreadsheet with raw data for all graphs.**
(XLSX)

**S1 Text. Supporting figures and tables. Fig A. Validation of the SigM-responsive reporter P(*amj*)-*yfp*.** (**A**) Representative fluorescence images of the indicated *B. subtilis* strains harboring the σ$^M$-responsive reporter P(*amj*)-*yfp* after exposure to the indicated antibiotics for 30 min. Scale bar indicates 2 μm. (**B**) Quantification of images from the strains in (**A**). Bar represents median. (**C**) Quantification of fluorescence intensity from images of wild-type *B. subtilis* harboring the σ$^M$-responsive reporter P(*amj*)-*yfp* after exposure to the indicated antibiotics for 30 or 60 min as indicated. Bar represents median. The lack of full activation of σ$^M$ by moenomycin is likely due to consumption of lipid II by the SEDS PG polymerases RodA and FtsW and generation of UndP. The data underlying B and C are provided in S1 Data. **Fig B. GFP-SigM membrane localization depends on YhdLK and rapidly relocalizes to the nucleoid when UndP is sequestered.** (**A**) Schematic of the reengineered SigM signaling system in strain BIR1100. A GFP-SigM fusion is expressed under the control of a xylose-regulated promoter. The SigM anti-sigma factors YhdL and YhdK are expressed under control of an IPTG-regulated promoter. The native *sigM-yhdL-yhdK* locus has been deleted (not shown). (**B**) Representative fluorescence images of the strain illustrated in (**A**). In the absence of inducers, there is faint GFP-SigM fluorescence. In the presence of 0.3 mM xylose, GFP-SigM localizes to the nucleoid. In the presence of both 0.3 mM xylose and 500 μM IPTG, GFP-SigM localizes to the membrane. (**C**) Representative time-lapse fluorescence images of the strain in (**A**) grown with 0.3 mM xylose and 500 μM IPTG before and after exposure to the indicated antibiotics. White carets highlight GFP-SigM localized to the nucleoid. Scale bars indicated 2 μm. **Fig C. Antibiotics that block UndP recycling rapidly deplete the free carrier lipid pool.** Representative images of wild-type cells treated with the indicated antibiotics for 2 min and then stained with fluorescently labeled MX2401 (MX2401-FL) and propidium iodide (PI). Staining was performed in the presence and absence of duramycin. Duramycin generates pores in the membrane, allowing MX2401-FL to access inward-facing UndP in addition to outward-facing molecules. Phase-contrast and PI staining highlight cells with permeabilized membranes. Scale bar indicates 2 μm. **Fig D. Inhibition of wall teichoic acid synthesis suppresses fosfomycin-induced SigM activation.** (**A**) Representative fluorescence images of wild-type *B. subtilis* cells harboring a σ$^M$-responsive reporter (P(*amj*)-*yfp*) after exposure to the indicated antibiotics for 30 or 60 min. The tunicamycin concentration (2 μg/mL) used in this experiment inhibits TagO, the committing enzyme in wall teichoic acid synthesis, but not MraY, an essential enzyme involved in PG precursor synthesis. Carets highlight morphological defects associated with inhibition of wall teichoic acid synthesis. Scale bar indicates 2 μm. (**B**) Growth curves of wild-type *B. subtilis* cells in the presence of the indicated concentrations of tunicamycin. Tunicamycin has not impact on growth at 2 μg/mL. The data underlying B are provided in S1 Data. **Fig E. Inhibition of wall teichoic acid synthesis suppresses the reduction in the free pool of UndP caused by fosfomycin.** Representative fluorescence images of wild-type *B. subtilis* cells treated with the indicated antibiotics for 20 min and then stained with fluorescently labeled MX2401 (M2401-FL) and propidium iodide (PI). Staining was performed in the presence of duramycin to generate pores in the membrane allowing MX2401-FL to access both inward- and outward-facing UndP. Merged phase-contrast and PI images highlight cells whose

membranes are permeable. The same MX2401-FL images are displayed in Fig 2C. Scale bar indicates 2 μm. **Fig F. Strains depleted of cell wall biogenesis factors have membrane integrity defects at the time point when SigM activity was analyzed.** Representative phase-contrast images overlaid with fluorescence from propidium iodide labeling. Each strain was imaged at the same time point as indicated in Fig 3C. Scale bar indicates 1 μm. **Fig G. Depletion strains have no morphological defects and do not activate SigM when grown under replete conditions.** (**A**) Representative fluorescence images of the indicated *B. subtilis* strains harboring the σ^M^-responsive reporter P(*amj*)-*yfp* grown in the presence of IPTG. Replete IPTG concentrations were 10 μM (*murAA*), 25 μM (*murB*), 25 μM (*mraY*), 12.5 μM (*murG*), 25 μM (*murJ*), and 25 μM (*bcrC*). (**B**) Overlays of phase-contrast and propidium iodide staining of the same images in (**A**). Strains display no morphological defects, have intact membranes, and do not induce SigM. Scale bar indicated 1 μm. **Fig H. Depletion of enzymes in the wall teichoic acid biosynthetic pathway that cause accumulation of UndP-linked intermediates activates SigM signaling.** (**A**) Schematic of the wall teichoic acid biosynthetic pathway. Enzymes depleted are shown in bold. (**B**) Representative fluorescence and phase-contrast images of the indicated *B. subtilis* depletion strains harboring the σ^M^-responsive reporter P(*amj*)-*yfp* after growth in the absence of IPTG. Inset highlights the morphological defects in the TagO depletion strain that do not cause SigM activation. Scale bars indicate 1 μm. (**C**) Growth curves of the indicated depletion strains grown in the presence (squares) or absence (circles) of IPTG. Red arrow indicates the time point at which samples were imaged. Permissive IPTG conditions were *tagO* (12.5 μM) and *tagG* (100 μM). (**D**) Quantification of YFP fluorescences from images as in (**B**). Bar represents median. The data underlying C and D are provided in S1 Data. **Fig I. Depletion of IspH activates SigM.** (**A**) Representative fluorescence and phase-contrast images of the indicated *B. subtilis* IspH depletion strain harboring the σ^M^-responsive reporter P(*amj*)-*yfp* grown in the presence or absence of IPTG. Even a partial depletion of IpsH (10 μM IPTG) causes SigM activation. (**B**) Growth curves of the depletion strain grown under replete (squares), partial depletion (triangles), or in the absence of IPTG (circles). Red arrow indicates the time point at which samples were imaged. (**C**) Growth curves of wild-type (WT) *B. subtilis* treated with fosmidomycin, vancomycin, or fosfomycin. Black arrow indicates the time when antibiotics were added. Red arrow indicates when sample was taken for imaging in Fig 4C. The data underlying B and C are provided in S1 Data. **Fig J. Sequestering UndP-linked sugars in cell surface glycosylation pathways activates SigM.** (**A**) Representative fluorescence images of the indicated *B. subtilis* strains harboring the σ^M^-responsive reporter P(*amj*)-*yfp*. Overexpression of YkcC or YkoT in the absence of YkcC or YkoT traps UndP-linked sugars and activates SigM. Carets highlight cells with morphological defects. Scale bar indicates 1 μm. (**B**) Schematic of cell surface glycosylation pathways. (**C**) Quantification of the YFP fluorescence in images similar to those in (**A**). Bar represents median. The partial suppression of SigM activation in cells pretreated with fosfomycin (fos) and tunicamycin (tunica) prior to IPTG addition is likely due to liberation of UndP from PG and WTA biogenesis pathways. The data underlying C are provided in S1 Data. **Fig K. Sequestering UndP in the teichuronic biosynthesis pathway activates SigM.** (**A**) Schematic model of the teichuronic acid biosynthesis pathway. The *tuaA* gene in PY79 is a pseudogene, and the committing step is thought to be catalyzed by TagO. The TuaB flippase is highlighted in bold. (**B**) Representative fluorescence and phase-contrast images of the indicated *B. subtilis* TuaB depletion strain harboring the σ^M^-responsive reporter P(*amj*)-*yfp*. Carets highlight cells with morphological defects. Scale bar indicates 1 μm. (**C**) Growth curves of the TuaB depletion strain grown in the presence (squares) or absence (circles) of IPTG. Red arrow indicates the time point at which samples were imaged. (**D**) Quantification of YFP fluorescence from images in (**B**). Bar represents median. The data underlying C and D are provided in S1 Data. **Fig L.**

**Schematic of the minor teichoic acid biosynthesis pathway.** Cells lacking GgaA exclusively make the major wall teichoic acid with polyglycerolphosphate. Cells lacking GgaB trap UndP in a minor teichoic acid precursor. **Fig M. Overexpression of UppS suppresses SigM activation caused by defects in the LTA synthesis pathway.** (**A**) Representative fluorescence images of the indicated *B. subtilis* strains harboring the σ$^M$-responsive reporter P(*amj*)-*yfp*. Strains were grown in defined rich medium with casein hydroyslate (CH) or in lysogeny broth (LB). The strains with an IPTG-regulated allele of *uppS* were grown in the presence of 500 μM IPTG. Scale bar indicates 1 μm. (**B**) Quantification of the YFP fluorescence from images like those in (**A**). Bar represents median. Overexpression of UppS in a wild-type background reduced SigM activity when grown in LB, consistent with UndP directly modulating SigM activity. The data underlying B are provided in S1 Data. **Fig N. Overexpression of UppS is toxic to *B. subtilis*.** (**A**) Representative fluorescence and phase- contrast images of the indicated *B. subtilis* UppS overexpression strain harboring cytoplasmic BFP. Cells overexpressing UppS (500 μM IPTG) are shorter and occasionally form mini-cells (white caret). Scale bar indicates 1 μm. (**B**) Growth curves of UppS overexpression strain grown with different concentrations of IPTG. Red arrow indicates the time point at which cells were imaged in (**A**). The data underlying B are provided in S1 Data. **Table A. Strains used in this study. Table B. Plasmids used in this study. Table C. Oligonucleotides used in this study.**
(PDF)

## Acknowledgments

We thank all members of the Bernhardt-Rudner supergroup for helpful advice, discussions, and encouragement; Tom Bernhardt, Suzanne Walker, and Andrew Kruse for insights and advice; Gen Dobihal and Xindan Wang for plasmids; and the MicRoN core for advice on microscopy.

## Author Contributions

**Conceptualization:** Ian J. Roney, David Z. Rudner.

**Data curation:** Ian J. Roney, David Z. Rudner.

**Formal analysis:** Ian J. Roney, David Z. Rudner.

**Funding acquisition:** David Z. Rudner.

**Investigation:** Ian J. Roney, David Z. Rudner.

**Methodology:** Ian J. Roney.

**Supervision:** David Z. Rudner.

**Writing – original draft:** Ian J. Roney, David Z. Rudner.

**Writing – review & editing:** Ian J. Roney, David Z. Rudner.

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
