## [Editor Report · Decision Letter 0]

6 Sep 2023

Dear Dr. Rudner, 

Thank you for submitting your manuscript entitled "Bacillus subtilis uses the SigM signaling pathway to prioritize its carrier lipid for cell wall synthesis" for consideration as a Research Article by PLOS Biology.

Your manuscript has now been evaluated by the PLOS Biology editorial staff and I am writing to let you know that we would like to send your submission out for external peer review.

Once your full submission is complete, your paper will undergo a series of checks in preparation for peer review. After your manuscript has passed the checks it will be sent out for review. To provide the metadata for your submission, please Login to Editorial Manager (https://www.editorialmanager.com/pbiology) within two working days, i.e. by Sep 08 2023 11:59PM.

Kind regards,

Paula

---

Senior Editor

PLOS Biology

---

## [Decision Letter · Decision Letter 1]

18 Oct 2023

Dear Dr. Rudner,

Thank you for your patience while your manuscript "Bacillus subtilis uses the SigM signaling pathway to prioritize its carrier lipid for cell wall synthesis" was peer-reviewed at PLOS Biology. It has now been evaluated by the PLOS Biology editors, an Academic Editor with relevant expertise, and by several independent reviewers. 

In light of the reviews, which you will find at the end of this email, we would like to invite you to revise the work to thoroughly address the reviewers' reports.

As you will see below, the reviewers agree that this is an interesting work, but further experimentation is needed. In particular, we think it is important to experimentally address the effect of overexpressing UppS, and, although we are aware that the mutant lacking TagO is not viable, the effect of lack of TagO if possible. 

Given the extent of revision needed, we cannot make a decision about publication until we have seen the revised manuscript and your response to the reviewers' comments. Your revised manuscript is likely to be sent for further evaluation by all or a subset of the reviewers.

**IMPORTANT - SUBMITTING YOUR REVISION**

*Re-submission Checklist*

*Published Peer Review*

*PLOS Data Policy*

*Blot and Gel Data Policy*

Sincerely,

Paula

---

Senior Editor

PLOS Biology

REVIEWS:

Reviewer #1: Jörg Stülke. Bacillus subtilis.

Reviewer #2: Bacterial surface.

Reviewer #1: Roney & Rudner provide a compelling in vivo analysis on the regulation of the YhdLK anti sigma factor via undecaprenylphosphate in B. subtilis. Even though the authors are unable to provide ultimate in vitro data (probably because of the problems intrinsically linked to the work with membrane proteins which preclude such experiments), the data are highly convincing. The paper is well written.

I have only very few comments:

1) l. 71: "Fosfomycin inhibits MurA": do you mean MurAA and MurAB or one of them?

2) l. 291 ff: The authors mention the broad conservation of sigM-YhdLK in Bacilli. However, there is no COG for YhdL and YhdK. YhdL is present in B. licheniformis, already the orthodox in B. anthracis has only very weak similarity (27.9% conservation), and the protein is absent from other important bacteria of the class Bacilli such as Listeria monocytogenes or Staphylococcus aureus. YhdK is present in B. licheniformis, but not even in B. anthracis not to mention other Bacilli. In conclusion I would use a statement like "SigM-YhdLK is confined to B. subtilis and close relatives of the genus Bacillus"

3) l. 311 ff: Given the absence of a strong conservation of YhdLK in pathogenic bacteria, I would not see a starting point for antibiotic discovery. Please delete this paragraph.

Reviewer #2: Bacterial surface glycans are assembled on a universal carrier lipid known as undecaprenyl phosphate (Und-P). Und-P is first synthesized as undecaprenyl pyrophosphate (Und-PP) by the UppS synthase. Und-PP is also released from Und-PP-linked intermediates during glycan polymerization. Und-PP is then dephosphorylated to Und-P by integral membrane phosphatases. At this point, the free pool of Und-P is distributed between peptidoglycan (PG) synthesis and multiple non-essential pathways. Thus, Und-P forms a vital regulatory nexus. However, despite its importance, virtually nothing is known about the mechanisms that distribute Und-P. Very little is even known about the events that occur when Und-P becomes limiting. Here, the authors show that in the Gram-positive bacterium Bacillus subtilis, disruptions in Und-P metabolism, either during de novo synthesis or in recycling, trigger signaling through the alternative sigma factor SigM. Briefly, the authors show that antibiotics that target Und-P, Und-PP, or block release of Und-P from glycans activate the SigM-responsive promoter PamJ. Similar effects were also observed for mutants that trap Und-P in dead-end intermediates. Reducing the rate of Und-P synthesis genetically or chemically also induced SigM signaling. Importantly, antibiotics or mutants not expected to reduce Und-P levels had little effect on SigM signaling. At this point, the authors worked to rule out the possibility that a sugar modification or secondary wall polymer was being sensed by SigM. The authors also showed that overexpressing the uppS synthase (increases Und-P levels) suppresses SigM activation in LTA synthesis mutants. Curiously, LTA synthesis is Und-P independent. Finally, the authors demonstrate that SigM becomes essential when Und-P metabolism is disrupted. This final result suggests that SigM functions to regulate Und-P distribution and has important implications for designing drug combinations.

Overall, the manuscript is well-written, the experiments controlled, and the conclusions mostly appropriate. The idea that SigM prioritizes Und-P distribution in Bacillus is an intriguing finding that should be followed up on. However, since the study does not include any direct Und-P measurements, the authors should include a little more data showing how overexpressing the Und-PP synthase (is known to increase the pool of Und-P) blunts SigM signaling. The authors also need to revisit the tunicamycin/fosfomycin experiment to rule out indirect effects. That said, the paper is nearly ready.

Antibiotics that block Und-P recycling rapidly activate SigM 

The authors show that amphomycin, bacitracin, vancomycin, and ramoplanin activate SigM signaling after 30 minutes exposure. Since these antibiotics are expected to disrupt Und-P recycling, the authors "support a model in which SigM is activated by a drop in the pool of UndP and a not a general block to cell wall synthesis." However, the authors do not measure the effect of these antibiotics on Und-P levels and SigM is still triggered (albeit at a lower level) by antibiotics not expected to disrupt Und-P metabolism (Figure S1C). For these reasons, the authors should determine if overexpressing uppS (like in Figure S10) delays/blunts SigM signaling for cells treated with antibiotics that disrupt Und-P recycling. 

Inhibition of WTA synthesis suppresses fosfomycin activation of SigM. 

The authors argue that fosfomycin, which does not directly induce Und-P sequestration, triggers SigM signaling due to the accumulation of Und-PP-linked WTA precursors (are attached to PG). To test this hypothesis, the authors simultaneously treated their cells with fosfomycin and tunicamycin, which inhibits the committing step in WTA synthesis. As expected, SigM activation was reduced. However, tunicamycin also inhibits MraY (PG synthesis). Thus, the fosfomycin/tunicamycin results likely represent a mixture of causes. Therefore, recommend testing the effect of fosfomycin on SigM activation in a mutant lacking TagO, which initiates WTA synthesis. 

Minor

Lines 14-15: "UndP is maintained at low levels in the cytoplasmic membrane (~105 UndP/molecules/cell)." Could the authors explain why they think 100,000 molecules is low? What level would be considered high?

Line 97: "tunicamycin reduced fosfomycin-mediated SigM activation at both 30 and 60 minutes (Fig. 2AB)." The authors should incorporate a concluding statement after this statement.

Lines 154-155: "the primary source of UndP for PG synthesis is the recycled lipid carrier and new UndP synthesis principally maintains the pool size during growth." Really? How would the cell distinguish between these two pools?

Lines 206-208: "Our model predicts that the overexpression of CsbB in the absence of YfhO activates SigM due to sequestration of UndP in UndP-GlcNac. If correct, the addition of tunicamycin and fosfomycin during overexpression of CsbB should have no impact on the activation of SigM." However, the authors contend that tunicamycin inhibits WTA synthesis (line 96). Since WTAs synthesis "requires a significant share the carrier lipid pool," inhibiting WTA synthesis would be expected to increase Und-P levels and potentially overcome the effects of Und-P sequestration.

Lines 237-241: This information should be included in the introduction, somewhere around Line 25, to reinforce the argument that SigM prioritizes Und-P for PG synthesis.

Figure 7B and 7C: The dilution factor should be noted on the figure or in the legend.

Is there a reason the authors did not include scale bars on their micrographs? 

Other

Figure 1A: "Antibiotics used in this study are shown in red." Should the authors include tunicamycin?

Figure S3: "Stains depleted" to "Strains depleted"

Line 91: "requires a significant share the carrier lipid" to "requires a significant share of the carrier lipid"

---

## [Editor Report · Decision Letter 2]

28 Feb 2024

Dear Dr Rudner,

My name is Luke Smith - I am an editor at PLOS Biology and have taken over the handling of your manuscript from my colleague Paula, who has recently left PLOS for a new job. Thank you for your patience while we considered your revised manuscript "Bacillus subtilis uses the SigM signaling pathway to prioritize its carrier lipid for cell wall synthesis" for publication as a Research Article at PLOS Biology. This revised version of your manuscript has been evaluated by the PLOS Biology editors and by the original Academic Editor. 

The Academic Editor is fully satisfied by the response to reviewers and based on this assessment, we are likely to accept your manuscript for publication. However, before we can editorially accept your study, we need you to address a few data and other policy-related requests, in a revision that we think will not take very long. 

**IMPORTANT: Please address the following editorial requests: 

1) TITLE: We would like to suggest a small tweak to the title, for clarity. If you agree, we suggest you change the title to: 

"Bacillus subtilis uses the SigM signaling pathway to prioritize the use of its lipid carrier for cell wall synthesis"

2) DATA: You may be aware of the PLOS Data Policy, which requires that all data be made available without restriction: http://journals.plos.org/plosbiology/s/data-availability. For more information, please also see this editorial: http://dx.doi.org/10.1371/journal.pbio.1001797

a. Supplementary files (e.g., excel). Please ensure that all data files are uploaded as 'Supporting Information' and are invariably referred to (in the manuscript, figure legends, and the Description field when uploading your files) using the following format verbatim: S1 Data, S2 Data, etc. Multiple panels of a single or even several figures can be included as multiple sheets in one excel file that is saved using exactly the following convention: S1_Data.xlsx (using an underscore).

b. Deposition in a publicly available repository. Please also provide the accession code or a reviewer link so that we may view your data before publication. 

>>Regardless of the method selected, please ensure that you provide the individual numerical values that underlie the summary data displayed in the following figure panels as they are essential for readers to assess your analysis and to reproduce it:

Fig 1C,D; Fig 2B; Fig 3B,C; Fig 4B,D; Fig 6B; Fig S1B,C; Fig S4B; Fig S8C,D; Fig S9B,C; Fig S10C; Fig S11C,D; Fig S13B; Fig S14B

>>Please also ensure that figure legends in your manuscript include information on where the underlying data can be found, and ensure your supplemental data file/s has a legend.

>>Please ensure that your Data Statement in the submission system accurately describes where your data can be found.

3) DATA NOT SHOWN: Please note that per journal policy, we do not allow the mention of "data not shown", "personal communication", "manuscript in preparation" or other references to data that is not publicly available or contained within this manuscript. Please either remove mention of these data or provide figures presenting the results and the data underlying the figures.

4) CODE: Per journal policy, if any code was generated to support the conclusions of your manuscript, we would require that you make it available without restrictions upon publication. Please ensure that any code is sufficiently well documented and reusable, and that your Data Statement in the Editorial Manager submission system accurately describes where your code can be found.

5) SUPPLEMENT: Please incorporate the supplemental methods into the main text

We expect to receive your revised manuscript within two weeks. 

*Published Peer Review History*

*Press*

Sincerely,

Luke

Lucas Smith, Ph.D.

Senior Editor

lsmith@plos.org

PLOS Biology

---

## [Editor Report · Decision Letter 3]

13 Mar 2024

Dear Dr Rudner,

Thank you for the submission of your revised Research Article "Bacillus subtilis uses the SigM signaling pathway to prioritize the use of its lipid carrier for cell wall synthesis" for publication in PLOS Biology and thank you for addressing our editorial requests in this revision. On behalf of my colleagues and the Academic Editor, Lotte Søgaard-Andersen, I am pleased to say that we can in principle accept your manuscript for publication, provided you address any remaining formatting and reporting issues. These will be detailed in an email you should receive within 2-3 business days from our colleagues in the journal operations team; no action is required from you until then. Please note that we will not be able to formally accept your manuscript and schedule it for publication until you have completed any requested changes.

PRESS

Sincerely, 

Luke

Lucas Smith, Ph.D.

Senior Editor

PLOS Biology

lsmith@plos.org